# Neutron Scattering at HIFAR—Glimpses of the Past

**Margaret Elcombe**

Australian Centre for Neutron Scattering, Australian Nuclear Science and Technology Organisation, New Illawarra Road, Lucas Heights, NSW 2234, Australia; mme@ansto.gov.au; Tel.: +61-2-9717-3611

**Abstract:** This article attempts to give a description of neutron scattering *down under* for close on forty-six years. The early years describe the fledgling group buying parts and cobbling instruments together to its emergence as a viable neutron scattering group with up to ten working instruments. The second section covers the consolidation of this group, despite tough higher level management. The Australian Science and Technology Council (ASTEC) enquiry in 1985 and the Government decision not to replace the HIgh Flux Australian Reactor (HIFAR), led to major expansion and upgrading of the existing neutron beam facilities during the 1990s. Finally, there were some smooth years of operation while other staff were preparing for the replacement reactor. It has concentrated on the instruments as they were built, modified, replaced with new ones, and upgraded at different times.

**Keywords:** neutron scattering; HIFAR; history

## 1. Prologue

On 30 January 2007, the Hon. Julie Bishop, Minister for Education, Science and Training initiated the final shutdown of Australia's first nuclear reactor, the 10-MW HIgh Flux Australian Reactor (HIFAR). Current and past staff of the Australian Nuclear Science and Technology Organisation (ANSTO) formerly the Australian Atomic Energy Commission (AAEC, or *the Commission*) and the Australian Institute of Nuclear Science and Engineering (AINSE) attended the special ceremony to say goodbye to an old friend. During its lifetime, this reactor has supplied millions of patients nuclear medicine doses (approximately half a million a year) to Australia, New Zealand and other countries in the Asia-Pacific region; has produced high grade Neutron-Transmutation-Doped (NTD) silicon for the semiconductor industry; and has allowed cutting edge neutron beam science to take place.

HIFAR went critical on Australia Day (26 January) 1958, was officially opened on 18 April 1958 by the then Prime Minister Robert Menzies, and was one of five carbon copies of the British designed reactor DIDO. In total, six DIDO class reactors were constructed based on this design:

- DIDO.
- PLUTO, also at Harwell, first criticality 1957.
- HIFAR (Australia), first criticality January 1958.
- Dounreay Materials Testing Reactor (DMTR) at Dounreay Nuclear Power Development Establishment in Scotland, first criticality May 1958.
- DR-3 at Risø National Laboratory (Denmark), first criticality January 1960.
- FRJ-II at Jülich Research Centre (Germany), first criticality 1962.

HIFAR was the last to shut down, in 2007. It was originally used as a materials-test reactor for the development of nuclear energy in Australia. However, in 1972, the Australian Government decided not to proceed down the nuclear energy route, and HIFAR was adapted for other purposes. The major uses were to produce radiopharmaceuticals, neutron beams for scientific research, NTD silicon and

radioisotopes for use in industry, for example gamma radiography to detect structural flaws in mechanical components and in tracing studies in the environment or through treatment plants. An immense amount of intellectual effort went into maintaining and developing HIFAR, to keep it running safely and effectively for the 49 years of its operating life.

When I arrived at the AAEC in 1967, neutron scattering had been gaining momentum for seven years. There were five operating neutron beam instruments, each with a single detector and no computer control. In the Australian tradition of ingenuity and few funds, two of these instruments were based on army-surplus gun mounts, and one had a car gearbox—a far cry from today's multidetector, highly computerised systems. None of those original instruments survived the introduction of computers and new technologies. However, the Triple-Axis Spectrometer, which I commissioned in 1971 and was revamped as a strain scanner, was still operating at shutdown.

The majority of users of the neutron beam instruments have been university researchers obtaining access via AINSE. Initially AINSE had its own scientific and technical staff who worked in parallel with the AAEC/ANSTO staff. This changed in 1991 when the instruments and scientific staff were transferred to ANSTO, while administration of proposals, travel funding and applications for government support for upgrading the neutron beam infrastructure remained with AINSE.

While preparing this paper, I have been privileged to be given unlimited access to the ANSTO library, HIFAR archives, and original instrument log books, as well as all the AINSE Annual reports and Council meeting notes. Finding out what took place between 1960 and my arrival in 1967 has taken many months. I have also had my own notes and many summary documents to draw upon to develop the overall picture of a continually striving group that often achieved much with very little, which, I believe, enabled it to progress to become the highly regarded Australian Centre for Neutron Scattering, ACNS, (formerly the Bragg Institute) that is in operation today.

Readers should also refer to AINSE's 40th anniversary book, which contains an article by Trevor Hicks [1]. This describes much of the early work from a university user's perspective and includes complementary pictures and comments to those I have included.

## 2. The Early Years—1960–1975

The first mention I can find related to neutron scattering in Australia is in the AAEC Open Day pamphlet for 1957 where a "Neutron Crystal Spectrometer" [2] (p.23) is briefly described (initially, all the instruments were labelled spectrometers, and the link between name and function came later). The following year at a United Nations (UN) conference in Geneva, in a paper describing the facilities being constructed at Lucas Heights,"Provision has been made for the subsequent installation of a velocity selector and neutron crystal spectrometers. One of the latter is in the process of manufacture" [3] (p.105). While the velocity selector was to be used for spectral measurement of neutron beams, a neutron crystal spectrometer would be one of the first neutron scattering instruments. Thus, neutron scattering *down under* was born.

While HIFAR and the Lucas Heights site infrastructure were under construction, research scientists were recruited and then seconded to nuclear establishments around the world to be trained for their future roles in Australia. Because HIFAR would be a carbon copy of DIDO, a reactor already operating at Harwell (UK), many Australian scientists were stationed there, including Terry Sabine who would lead the neutron scattering group in its early years [4].

The first neutron beam in-pile collimators (also copied from DIDO, but manufactured in Australia) were installed into 4H1 and 4H2 on 16 June 1959 (refer to Table 1 for names and locations of instruments over time (HIFAR has 10 faces numbered in an anticlockwise direction starting opposite the control room with Face 1 the thermal column; a layout diagram of the status in 1998 is included in [5])). The 4H2 collimator was rather a tight fit and its removal caused significant delays to the upgrade program of 1989–1991. However, it did mean that the internal dimensions of all the remaining holes were accurately measured so this problem should not arise again. There were also issues with the removal of the 6HGR9 Collimator (X78) installed on 10 October 1960, but not nearly as serious.

The first instrument, a long wavelength spectrometer, was installed on 4H1 in April 1960, but I think it was never used as a spectrometer there. This was a long wavelength spectrometer. It included a liquid nitrogen cooled Be/BeO/Bi filter 10–14 inches long, a beam diffracted by a mica crystal and a moveable counter arm. The beam monitor was a FEU2/90 2 inch diameter fission counter and the detector was $BF_3$ (2 inch diameter, 12 inch long). The detector could be positioned from $5°$ in intervals of $5°$. Samples $\frac{3}{4}$ inch in diameter and 6 inches long were measured for a pre-set monitor count and then the standard was measured for the same monitor count. The filter was cooled automatically using $\approx 10$ L liquid $N_2$/day. It was only intended to be at this location for 4–5 months. In May 1960, it was proposed to remove the filter to measure the thermal spectrum from the in-pile collimator. Radiation surveys and spectrum details are reported in various HIFAR Technical Notes.

**Table 1.** Instrument Locations over Time.

| Face | Identifier | Year | Collimator | Instrument(s) |
|---|---|---|---|---|
| 2 | 6H | 1962 | X41 single beam | General Purpose Instrument and SCS |
|  |  | 1963 |  | Triple Axis 0 |
|  |  | 1966 |  | SCS |
|  |  | 1968 | X41/2 twin beam | Triple Axis 1 and AAEC (CCD) |
|  |  | 1979 |  | HRPD and AAEC CCD |
|  |  | 1991 | X213 Helium filled | AUSANS |
| 3/4 | 2 Tan A and B | 1961 | X82 | AINSE SCS |
|  |  | 1963 | X82/2 | AINSE SCS |
|  |  | 1965/6 |  | Vacant |
|  |  | 1967 | X82/4 | AINSE "Mangrovite" SCS |
|  |  | 1969 |  | Additional Manual SCD |
| 5 | 4H1 (upper) | 1959/60 | X2 | Long wavelength filter spectrometer |
|  |  | 1961 |  | Powder instrument |
|  |  | 1991 | X206/1 | Medium Resolution Powder Diffractometer |
| 5 | 4H2 (lower) | 1959/60 | X3 | AAEC SCS |
|  |  | 1991 | X206/2 | Relocated HRPD |
| 7 | 10H | 1962 | X48 |  |
|  |  | 1969 | X48/3 | Triple Axis Spectrometer |
|  |  | 2003 |  | Converted to Strain Scanner |
| 8 | 6HGR9 | 1960/1 |  | Long wavelength mechanical monochromator |
|  |  | 1971 | X78 | AINSE LONGPOL test setup |
| 8/9 | 2 Tan X | 1968 | X166 source block | Small Angle Scattering Instrument |
|  |  | 1971 |  | Various experimental setups interferometry, Laue etc. |
| 9 | 6HGR10 | 1969 |  |  |
|  |  | 1976 | X172 | LONGPOL-1 |
|  |  | 1989 |  | Intermediate-Q instrument test bed for area detector. |
|  |  | 2003 |  | Reflectometer |
| 10 | 4H5 | 1960 | X21 | Long wavelength filter spectrometer |
|  |  | 1964 |  | AINSE SCS (from 2 TanA) |
|  |  | 1970 | X21/2 |  |
|  |  | 1971 |  | Powder spectrometer and polarised neutron spectrometer |
|  |  | 1987 |  | Biological membrane instrument and LONGPOL-2 |

SCS: Single Crystal Spectrometer; AAEC: Australian Atomic Energy Commission; CCD: Computer Controlled Diffractometer; HRPD: High Resolution Powder Diffractometer; AUSANS: Australian Small Angle Neutron Scattering Instrument; AINSE: Australian Institute of Nuclear Science and Engineering.

The instrument is shown in the 1960-61 AAEC Annual Report [6] located on the 4H5 hole so it was probably moved there shortly after a collimator was installed on 13 October 1960 (Figure 1). The first scientific publication from a HIFAR neutron beam was from this instrument [7]. That paper in Nature states that the spectrometer was similar to that described by Antal and Goland [8].

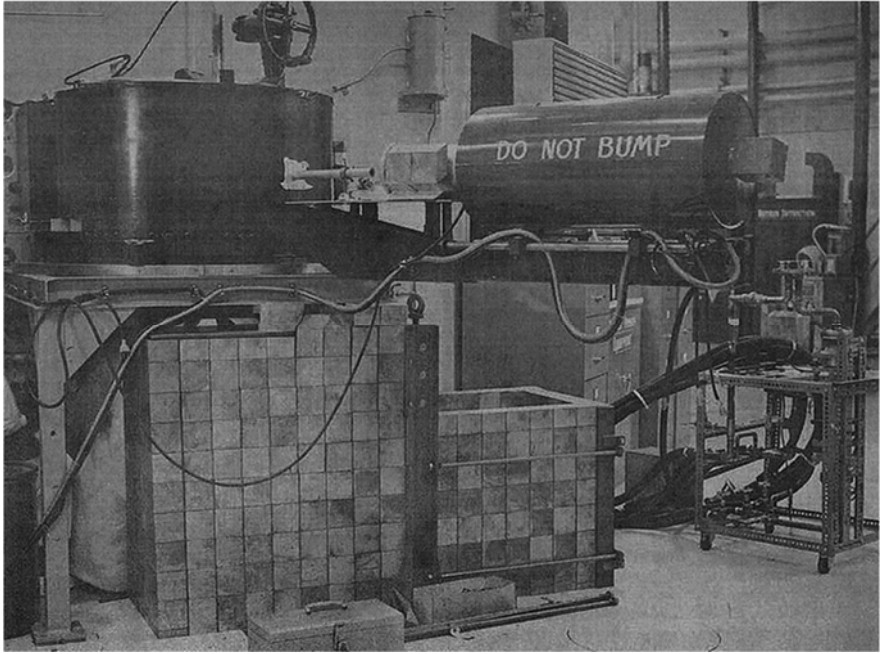

**Figure 1.** Long Wavelength Neutron Spectrometer on 4H5 circa 1960. Australian Atomic Energy Commission Annual Report 1960-61 (p. 75); © ANSTO used with permission.

The next instrument was the AAEC Crystal Spectrometer on the 4H2 hole, probably installed in late 1960 (It is shown as *existing* in Drawing AE8526 (8 December 1960)). The initial shields were unsatisfactory (too small and water filled) and an early photo shows additional shielding packed around it. Permanent shields were installed in May 1961. The spectrometer was put to use immediately and the first paper from it was only a month behind the Nature paper mentioned above [9]. Data collection from this was rudimentary. There was a coupled $\Omega/2\theta$ drive and a $\phi$ circle but no $\chi$ variable so the sample was mounted with a zone axis vertical. The data were collected by a series 150 control unit [10] and output as a list of numbers on a piece of paper (about 40 counts an hour). Unfortunately, the starting positions of each scan had to be set manually, so continual attendance was required and long shifts involved as noted by Hugo Rietveld, who was a PhD student from University of Western Australia at the time [11]. The 2-D data was combined with 3-D X-ray data to confirm the structure of *p*-diphenylbenzene. The Fourier plot from the neutron data is given in the paper. The full structural paper [12] is one of the highly cited papers from HIFAR (Appendix A).

What was to become the AINSE powder spectrometer was under construction at this time (by the AAEC). The original design was modified when two army-surplus gun mounts were purchased (£25 each I believe, not uncommon for neutron scattering instruments). These made ideal $2\theta$ drives, one for 4H1 and the second for the general purpose spectrometer on 6H. On 4H1, the base was driven through a Holden car gearbox and could drive smoothly either up or down at a fixed speed. The first data were displayed on a chart recorder and the intensities estimated from the area under the peaks. One could hardly call it measuring, as I believe cutting and weighing the paper was involved on some occasions. Although it was built by the AAEC, it was donated to AINSE in 1965 together with a liquid-helium cryostat. A water-cooled electromagnet followed a year later. By this time, it was controlled by a series 150 unit. There are several early published papers from it [13–16].

The first instrument purchased by AINSE was a single crystal spectrometer, which was installed on 2TanA. There were some problems with this instrument. Initially, its shielding was inadequate (October 1961), so an extra semicircular shield was made to fit around it together with a substantial lid (December 1961). However, although several university users claimed to have collected useful datasets from the instrument, no early publications resulted. The beam had a poor profile, sometimes referred to

as a split beam, which was not rectified until a water source block was installed underneath the reactor core from the opposite end of the facility (May 1968). However, two 2-D datasets collected from it in 1962 were used. The structure of rubidium uranyl nitrate was published in 1965 [17]. From a starting structure determined by X-rays and dominated by the heavy atoms, difference Fourier maps were used to locate the N and O atoms in the 0K.L (triple hexagonal cell indices) and hk0 (rhombohedral indices) planes. It seems that the data (even if poor quality) were ahead of the computing programs. An AINSE post-doctoral fellow, working on the computer programs needed for crystal structure solution, reported both computing and structure solution problems [18]. Because of the beam profile problems, the instrument was moved (early 1965) into the position previously occupied by the long wavelength spectrometer on 4H5. Papers from the instrument in the new location started in 1967.

This movement of instruments (or parts of instruments) was quite common in those early years. Although it is not always clear from the references, the few photographs that remain from that time clearly show that this occurred [1,19–23]. Sabine 1964 [22] includes photos of five instruments: the pair on 4H1/4H2, powder spectrometer above and single crystal spectrometer below, a General Purpose instrument (set up as a Triple Axis Spectrometer) on 6H using the second gun mount as the spectrometer base (Figure 2), the long wavelength neutron spectrometer on 4H5, and the mechanical monochromator on X78. This last instrument was being developed to replace the long wavelength neutron spectrometer. The instruments now had a common data collection protocol. The detector accumulated counts for a pre-set number of monitor counts, the count was output onto paper tape, the motor was driven a pre-set number of steps and the process repeated until the instrument was stopped. This process was controlled through an AAEC series 150 digital counting unit [10].

The General Purpose Spectrometer on 6H has an interesting history. It started out as a powder instrument with a gun mount as its base. It was also possible to attach a Stoë χ-circle to it and perform single crystal measurements. Trevor Hicks suggested that it could be turned into a triple axis by adding a second θ/2θ drive onto its long counter arm and relocating the counter onto the new 2θ base. I am not aware of any published inelastic work from this configuration. I doubt whether the series 150 driving/counting system could cope with the extra driving channels.

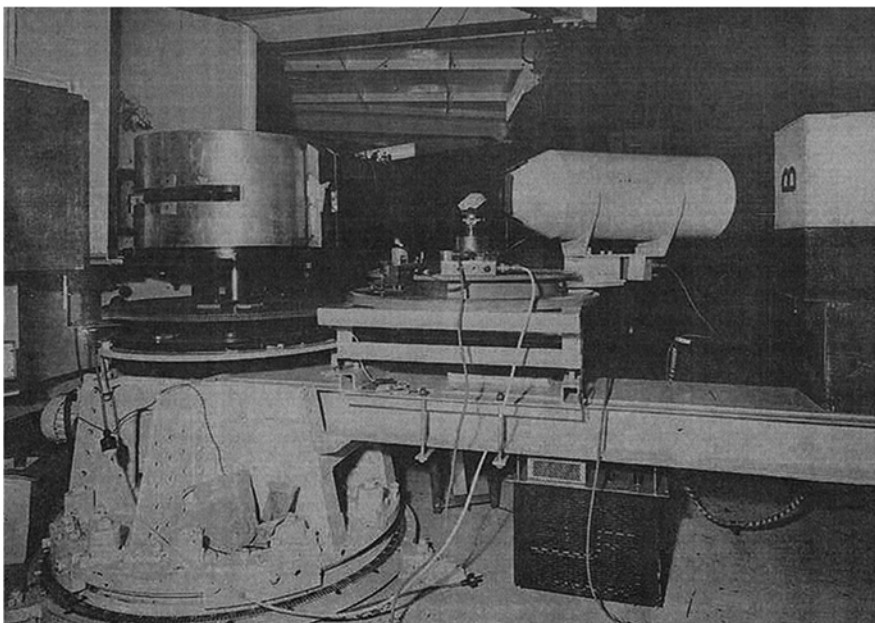

**Figure 2.** General Purpose Spectrometer on 6H as a Triple Axis Spectrometer circa 1963. AAEC Staff Reprints 1955-1985 #321 (Figure 4, p. 10); © ANSTO used with permission.

It was moved out of position for a year or so and the θ/2θ section with a smaller counter shield was mounted on a fixed frame close to the shield to form a single crystal spectrometer. With a tiny cryostat mounted inside the ring of the Stoë χ-circle, this configuration was used to collect room and low temperature data from thiourea and deuterated thiourea leading to a classic (and highly cited) ferroelectricity paper [24]. Within the cryostat the crystal was mounted with a 110 zone axis vertical so complete 3-D data were possible with the χ-circle restricted to $\pm 50°$ tilt. The cryostat had to be topped up with liquid nitrogen every 20 min for a week and long shift-work hours were involved. The instrument also had to be positioned manually for each reflection. This instrument was used extensively for ≈1 year. After that, the in-pile collimator was replaced by a twin beam unit. The single crystal spectrometer was relocated onto the *A* beam and the rewired triple axis was reinstated on the *B* beam, with a track on the floor to take the extra load as the counter shield had been increased by a *wooden coffin surround* and a large lead counterweight added. It was now possible to step the sample, scattering and analyser angles independently so that constant-Q measurements (albeit linearised) were possible. Unfortunately, starting positions for each scan still required manual setting (not nearly as arduous a task as for the single crystal instruments as inelastic measurements are notoriously slow). The first reported phonon measurements were from zinc [25], calcium fluoride [26] (highly cited) and strontium fluoride [27]. Once the new Triple Axis Spectrometer was commissioned (see below) this instrument reverted to an elastic powder diffractometer, which with the addition of a rotating liquid nitrogen cryostat, became a workhorse for John Taylor and Paul Wilson who solved the structures of many uranium compounds. This work often used X-ray data (either single crystal or $F^2$ extracted from powder patterns) to determine the uranium positions and suggest light atom locations. Complete structures were refined by profile analysis of the neutron patterns [28]. The standard least squares program of Busing Martin and Levy [29] was modified in-house for "the direct fitting of structures to powder diffraction patterns by the method of profile analysis" [30]. Multiphase refinement was first accomplished several years later [31].

The AAEC single crystal spectrometer, now repositioned on 6HA, and the AINSE 2TanA instrument (a new instrument installed in 1967 to replace the original), were adapted for computer control using a single PDP8 computer [32]. About this time AINSE purchased a third single crystal instrument that was set up on the 2TanB beam line. This instrument was still manually operated and did not see much use. The opposite end of the tangential facility X166 was set up with a very narrow collimation to feed a small angle scattering instrument. Again, this was more of an experimental instrument, which was set up to suit the needs of specific users as they arose.

The long wavelength mechanical monochromator instrument (also known as X78) had a liquid nitrogen cooled beryllium filter, a mechanical chopper with rotation axis set at ≈4° to the beam direction which controlled the wavelength between 5 and 12 Å by varying its speed of rotation, a sample positioner and a well shielded detector. Once the work on Be and BeO was finished it was used for irradiation damage studies of MgO [33]. By the early 1970s, this work was also finished and the first version of LONGPOL was being developed to use the long wavelength beam for a wide range of magnetic studies [34,35]. To achieve both increased intensity and optimum polarisation, the angle offset was reduced to 2°35′ to allow 3.6 Å wavelength to be used.

While all the instruments so far were constructed from purchased components and home made parts, the Triple Axis Spectrometer (TAS) using a concept design by Arthur Pryor was designed in detail by Weapons Research Establishment in SA and manufactured by Vickers Ruwolt in Melbourne. It started arriving in 1969 and was installed and commissioned on the 10H hole during 1970/1971. Compared with earlier instruments, it was huge (22 tonnes) with the single rotating monochromator shield weighing 8 tonnes. All its components neatly wrapped into each other and, with a large diameter detector shield, was renowned for its low background. It was fully computerised with all six motors and angle encoders plus counters controlled by a single PDP8 computer (4 Kb total size). It was capable of both constant-Q and constant-E scans.

Once the single crystal spectrometer was working well on 2TanA, AINSE got to work on the 4H5 beams [36]. The 10-inch-by-4-inch hole allowed two beams to be extracted independently, one of which had three possible collimations. A new densified wood (Wodasteen) was used for the monochromator shield. The side with variable collimation was set up as a second powder diffractometer. The other one, anticipating a variety of future user requirements, was developed as a single crystal instrument with a tilting counter. This would allow data collection from a sample mounted in a cryostat without having to tilt the cryostat. A polarising CoFe monochromator was available for magnetic structure studies. The instrument was strong enough to hold an electromagnet around the sample position.

Many believe that only the reactor operators and a few staff were allowed into the HIFAR containment building. However, during the Open Days in 1969 and 1971, groups of visitors were escorted through HIFAR and the list of neutron scattering instruments for both dates is given. Below is the extract from the 1971 booklet [37] (p.41). Part of route through HIFAR building (entry through B40 airlock exit through B42 airlock):

*Neutron Diffraction Experiments.*

*Neutron beams are brought out from HIFAR through narrow holes in the shielding called collimators. They are directed onto specimens which scatter the neutrons and the scattering is recorded by neutron counters. The technique gives information about the atomic structure of the materials.*

*As you pass around the faces of the reactor you will see in order:*

(a) *Variable Wavelength Triple Axis Spectrometer used for studying the vibrations of atoms in crystals.*

(b) *Long Wavelength Neutron Scattering Apparatus for studying irradiation induced defects in materials.*

(c) *Small Angle Scattering Apparatus for studying magnetic interactions in materials.*

(d) *Polarised Beam Spectrometer for determining the magnetic structure of crystals.*

(e) *Powder Spectrometer used for studying the atomic and magnetic structures of materials which do not form good single crystals.*

(f) *Computer Controlled Diffractometer used for determining the atomic structure of crystals.*

(g) *Fixed Wavelength Triple Axis Spectrometer—use similar to (a).*

(h) *Computer Controlled Diffractometer used as for (f).*

(i) *Manual Diffractometer used as for (f).*

(j) *Computer Controlled Diffractometer used as for (f).*

(k) *Powder Diffractometer used as for (e).*

*Opposite (d) there is a display bench showing:*

(i) *Breakdown of shield construction.*

(ii) *Measurement of typical temperatures used.*

(iii) *Demonstration of magnetic field strength.*

(iv) *Typical crystals used in neutron scattering experiments.*

Simple placards, not fancy colourful posters as expected nowadays, were attached to the above instruments explaining their use and these are given in Appendix B.

Initially, as already mentioned, the instruments were set manually by the instrument scientist collecting the data. After initial test runs by the research scientists, this task was added to the duties of the *rig technicians* who were part of the reactor operating team. They were supplied with computer printouts of (*hkl*) and angles [38] and given the hourly task of resetting the instrument to collect a fresh reflection. This greatly relieved the stress on the scientists but the data were still unreliable. Mis-setting of angles was quite common, particularly in the small hours of the morning. This was made clear to me by one scientist who deleted all data collected between 5:00 a.m. and 7:00 a.m. and gained a substantial

improvement in the quality of the structural refinement. Once the single crystal instruments were computer controlled, both the speed and reliability of data improved markedly. Many ingenious pieces of code were developed:

- A simple instruction allowed data collection from an octant (for example) within a given 2θ range.
- The axes had degree markers attached, checking every revolution to 0.01°. If it were only out by a few hundredths, it would reset that axis and continue. If the error were significant, it would stop collecting data, re-zero all axes, recollect its standard reflection and then go back to where it had left off. This ensured that all the data collected were at the correct angles. Slow data collection, or a change in the standard reflection intensity, meant that a motor drive was failing and maintenance was required.
- Data were collected not as a single count but as four quarter counts. Each quarter was compared to the previous one or the running average and if the difference was greater than allowed for by Poisson statistics, the counts were rejected and that count restarted. This ensured that extraneous noisy counts were excluded. If data collection became slow, it was a sign to the user that something was at fault with the instrument and help should be sought. In the long run, this saved time as in the past it was only during processing and refinement that data problems were detected.
- Time was not expended collecting background for long times when the diffraction peaks were strong. Single counts at peak and at background were used to determine the time for background collection such that its contribution to the statistics was comparable to that of the peak.
- Regular collection (every 25 intensities for example) of up to three standard reflections ensured that any movement of the crystal could be picked up early, and the alignment matrix reset.

In the manual days, the data had been output onto five-hole paper tape as a long string of individual counts (preceded by identifiers including reflection indices). A suite of programs had been developed over time to extract the integrated peak intensities from such data and prepare them for input into phasing programs, a Fourier program and eventually structure refinement programs. Now the data were output as integrated intensities with *hkl*, 2θ and peak position error onto computer tape (eight-hole) by the teletype. All the mainframe programs were updated for the new input data format [39], and outputs now made available in many formats suitable for the users' ongoing structure solution programs.

This period could be described as a *golden period* when, in some cases, Lucas Heights led the world. Mini-computer control of instruments was developed at Lucas Heights [32] before it was used at Harwell, or even at the US Brookhaven and Oak Ridge laboratories. The strategy of improving data reliability [39] was an important innovation. Lucas Heights was the first laboratory anywhere to encourage university users, and Hugo Rietveld did his first experiments with Terry Sabine. This resulted in the establishment of AINSE in 1959, only copied in the 1970s at Harwell and later in Europe and the US. This university support proved vital in the tough times to come, and today most of the world's neutron scattering involves university users.

Thus, by the early 1970s, neutron scattering in Australia was established, was well on the way to having 10 operational instruments at HIFAR and publications were being produced at about 14 per year. The publications/year (however unreliable) are shown in Figure 3. The future looked good.

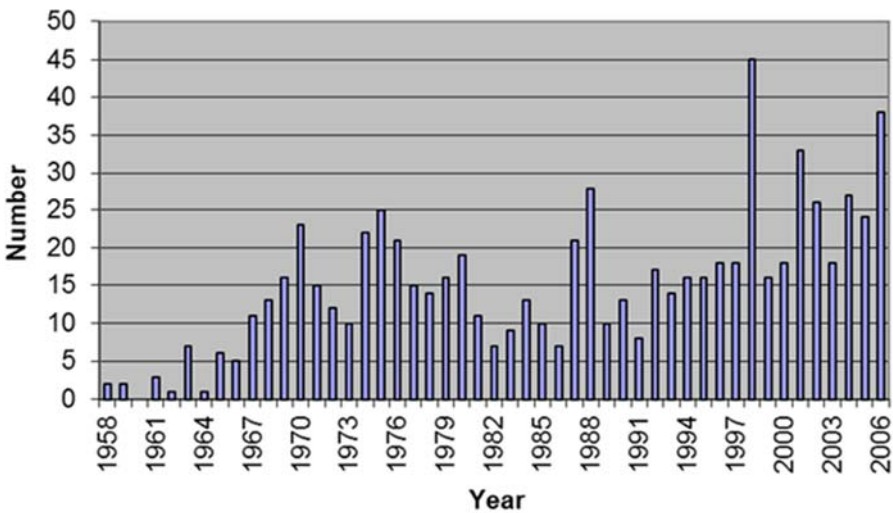

**Figure 3.** Neutron scattering publications using the HIgh Flux Australian Reactor (HIFAR) 1958–2006. The publications from HIFAR consist of a mix of listings. It was started by the late Lindsay Davis for the first Australian Neutron Beam Users Group (ANBUG) submission in 1981. He maintained and updated it with every new submission until 1992. I took this over (unfortunately just a hard copy listing) and kept it updated through all the grant applications during the 1990s (as a mix of word documents) and then supplemented it using an AINSE EXCEL spread sheet up to 2005, and the Bragg Institute database for 2004, 2005 and 2006. When determining the citations, I found errors, omissions and duplications in the listings. It may not be accurate, but it is the best there is. (HIFAR Publications, Australian Centre for Neutron Scattering, HIFAR Archives; © ANSTO used with permission).

## 3. Consolidation 1975–1985

There then followed a period of consolidation and adaption to user demand. The increase in the speed of data collection brought about by computer control of the single crystal instruments meant that we did not need five such instruments. The AAEC instrument on 4H2 fell into disuse and 2TanB was only used occasionally when it was set up with special user requirements.

The LONGPOL instrument on X78 had proved the polarisation analysis concept envisaged by Trevor Hicks but lacked neutron intensity. As no money was available for a new shield, an old shield was located in the transit store, modified to take a dog-leg monochromator to both lower the beam and allow for a variation in wavelength, and positioned on X172. LONGPOL was moved there with some minor improvements and worked well for a number of years. By late 1981, the use of the polarised neutron spectrometer (on 4H5) was declining and a decision was made to design a new LONGPOL to use that beam while retaining the low angle biological instrument on its existing beam, but with an increased take-off angle. The design was radically different in that the whole instrument was to be set up *inside* its own shield rather than shield each individual component. Most of the internal components were designed and constructed at Monash University (Melbourne, Australia). It was installed in late 1987, with the original 40% efficient iron polariser and analyser system.

The use of the powder instruments was limited to some extent by their resolution (good at low angles but very poor at high angles). They were very good for magnetic studies, where the intensity is largest at low angles, and for determining phase transition temperatures when used with variable temperature cryostats. However, with the design [40], construction and demonstration [41] of the high resolution diffractometer D1A at the Institut Laue-Langevin (ILL), France and the availability of a profile refinement program [42], these early restrictions could be largely overcome. This immediately

led to the request that the 6HB powder instrument be replaced by a high resolution diffractometer. Prof T.M. Sabine and Dr J.C. Taylor used their powers of persuasion to help get the project under way. A new shield was developed to suit the existing in-pile collimator, maintain the 6HA single crystal instrument position, allow for both high and medium resolution options with the same spectrometer position, and allow for further improvements using in-shield soller collimators. The original spectrometer was repositioned and, with a high resolution soller collimator positioned in front of the detector, a working single detector high resolution powder diffractometer (HRPD) was ready by the end of 1979 [43]. The detector shielding was modified to take additional detectors, each with its own soller collimator, and eight detector channels were established in 1983. It operated successfully until 1991 when it was relocated as part of a further instrument upgrade (see below). It was the first instrument at the AAEC to have a PC to control it—a MINC11 using eight-inch floppy disks for transporting data. Programming this was a much simpler task than the PDP8s. In parallel with the instrument development, and based on a program (DBW3.2) of Wiles and Young [44], a new refinement program using the Rietveld technique was developed jointly by the AAEC and the Commonwealth Scientific and Industrial Research Organisation (CSIRO) (LHPM1) [45] and the successful demonstration of quantitative phase analysis [46] produced the most highly cited paper from HIFAR. While the powder instrument was being upgraded the opportunity was taken to install a primary beam collimator and new monochromator onto the single crystal diffractometer (6HA). As a result, good quality data were obtainable, albeit slowly.

The Triple Axis Spectrometer was fully operational by mid-1971 and its accuracy confirmed by measurements of the phonons in nickel. However the earliest published inelastic measurements I can find are in 1983 [47,48]. The instrument had several engineering hiccups.

- The first time we lifted the monochromator plug to change the crystal a very high radiation field was detected. The depleted uranium shield attached to the plug had not been protected from the neutron beam. With allowances for radiation cooling, obtaining special permission and conditions to trim the uranium back, and then applying the $B_4C$ araldite neutron absorber, took time. A reassessment of all parts of the shield which could be exposed to the neutrons was undertaken and boroflex sheet was applied wherever possible.
- After the first extended reactor shut down since the instrument was commissioned the recalibration showed a $2°$ shift in $\phi_o$. This was eventually traced back to a drive mechanism which was slightly different in the positive and negative directions $\approx 0.01°$ per $\pm 90°$ drive. It was small and always in the same direction so it had mounted up over two years. Monthly recalibration was necessary from then on.
- In mid-1973, the neutron flux started declining and looked set to fail in about six months. This was traced to a flaw in the in-pile collimator design. The input and output of the helium cooling circuit were at the same end of the inner tube. Each reactor shut down the whole system cooled and a small amount of air + water vapour was sucked into the helium space, where the water condensed. Each start up some water vapour remained in the collimator tip and absorbed neutrons. The system was flushed well with dry nitrogen gas, one of the helium lines was diverted to the spare line which provided better access to the inner tube and from then on, a low positive helium pressure was maintained into the collimator to prevent a recurrence. Three months later, the neutron flux recovered to almost its previous value.

To get the best out of the instrument, the monochromator and analyser needed to be matched, so a program was put in place to develop improved monochromators mainly by increasing the mosaic spread by mechanical means. The copper monochromators were improved by cold rolling but they still had the problem of higher order contamination. Hot pressing of germanium was being done overseas and this was attempted in Australia. Eventually we started getting good results. Throughout the 1980s, the triple axis ran well with many publications resulting. It was not always used in its intended fashion. Beam energy characterisation and the successful measurement of $\bar{\nu}$ for plutonium [49], successful

(and I believe first) observation of the neutron Kikuchi effect in lead [50] and diffuse scattering from cubic stabilised zirconia [51] were some of these.

Over this period the demand for neutron scattering came almost entirely from the university sector. This was all supported through AINSE which through specialist committees selected the proposals to be supported and provided travel and accommodation grants to users to come and stay at Lucas Heights. It also provided in-house scientific and technical support to the users both while collecting the data and with ongoing support while the data were being analysed and published.

At the same time, the AAEC support for neutron scattering was in decline. Staff left or moved to other areas and were not replaced. With the HRPD coming on-line, which was only possible because of substantial AINSE support, it was realised that more AAEC support was needed. A case was prepared for senior management in mid-1980 recommending a substantial change in the role of the neutron scattering group and that ≈$3.1 million be spent to upgrade the current facilities including substantial effort in neutron detection and data handling systems. It was recommended that the neutron scattering group be transferred to the Applied Physics Division of the AAEC at the end of that year and a rebuild commence.

During this period the international scene was changing. New reactors were coming on line specifically built for neutron beam science with tangential holes giving a much lower fast neutron background. Maybe the radial holes from HIFAR were not ideal for neutron scattering. The near failure of the triple axis beam and other reactor problems had management worried that HIFAR might not last as long as originally anticipated. This must have filtered out as a paper was prepared by AINSE [52] detailing the *Neutron Beam Needs on New Reactor* which included thermal flux, cold source, neutron guides, collimators, building and space requirements. Although this did not lead anywhere directly it was not long after this that the Australian Neutron Beam Users Group (ANBUG) was formed. This group was mainly composed of external researchers who could lobby management and government on behalf of the users to improve the facilities. The first major proposal prepared by the group in 1981 was for a cold neutron source to be inserted in the 6H collimator hole [53]. Cold sources had already been successfully inserted this way into similar reactors overseas. The longer neutron wavelengths that such a cold source would provide were essential in order to keep up with the developments in soft matter science occurring elsewhere. Another paper in 1984 [54] was a submission to the Australian Science and Technology Council (ASTEC) and included among other recommendations that staff numbers be increased from the current level of 8 to 14 (to return the staffing level and staff to instrument ratio to the 1972 values) to make the AAEC comparable with similar organisations overseas. The AAEC was also continually updating the justification and need for a new reactor. The outcome of the National Energy Research Development and Demonstration Council (NERDDC) review in 1981 [55] led to approval in 1982 of a refurbishment of HIFAR to ensure it would run safely until at least 1990.

This continual push for improved facilities led to a major government enquiry which resulted in the ASTEC report tabled in the House of Representatives on 28 November 1985 [56]. This recommended ≈$10 million be spent over the next 10 years "to maintain and upgrade HIFAR so that its continued safe operation can be guaranteed and improvements be made to the neutron beam instrumentation; this will ensure a continued Australian capability in neutron beam research and applications and radioisotope production" [56] (p.4). In addition, it recommended that travel funds ≈$250 thousand be made available through the Australian Research Grants Scheme "to be used to support the travel of Australian scientists whose projects have been determined to be of the highest international standard and require accelerator or neutron beam facilities which are not available in Australia" [56] (p.5).

The ongoing lack of a cold source meant that Australian scientists working in the field of soft matter science, for example, could not perform their experiments within Australia. Some Australian scientists became known for *suitcase science*, becoming well known at neutron centres around the world, performing experiments, usually in collaboration with local scientists, using techniques such as

small angle neutron scattering (SANS) and reflectometry, on instruments supplied with cold neutrons. However, their welcome was becoming thin as the demand for the overseas facilities increased.

In Australia, the early 1980s was a time of steady use of seven neutron scattering instruments: two powder diffractometers, two single crystal diffractometers, a triple axis spectrometer, LONGPOL and a low angle diffractometer for membrane studies. With continued AINSE encouragement, by 1985, there were plans in place to install three more computers, to provide an absolute angle scale onto 4H1 with three detectors, and upgrade the TAS computer from the original PDP8 (4 Kb), which had become somewhat unreliable, to a PDP11/23 (256 Kb). Although this new computer came with a Fortran compiler, the operating system of the instrument was essentially unchanged, and some sections of code had still to be written in machine language to make this possible. The publication rate appears to have declined from a peak rate of ≈20 papers per year in the early 1970s to ≈10 papers per year (see Figure 3). However, there was a steady increase in publications by Australian scientists using overseas facilities and quality may have won over quantity as of the highly cited papers from HIFAR (Appendix A) about half come from the 1980s.

## 4. Expansion 1985–1999

As a result of the ASTEC recommendations discussions were started about how best to upgrade the neutron scattering instruments. A paper was presented to the Commission in December 1986 with options ranging from $2.6 million to $15.5 million (the expensive option included a cold neutron source, guides, an experimental hall and four new instruments). The Commission accepted the $2.6 million option for a Small Angle Neutron Scattering Instrument and modernisation of other instruments and would pursue government appropriation of $300 thousand for 1887/1988. This looked promising. However, in 1988 the AAEC Act was amended and the Organisation was revamped with a new mission as the Australian Nuclear Science and Technology Organisation. The Government also required that the Organisation bring in real revenue to supplement the appropriation funds. If the revenue targets were not met, then staff numbers might be reduced. Then followed the start of applications to various funding bodies for collaborative research grants (with universities and/or AINSE) to effectively get money for staff (an experimental officer for three years and a research fellow for two years). Research Infrastructure grant monies were counted as revenue. I am only aware of three real commercial jobs, each worth <$10 thousand.

It was possible to make modest improvements in-house. Some examples were

- Two furnaces, the P1100 (1991) and P1700, were designed in-house and manufactured locally and provided good service.
- A robust, air activated 30 position sample changer was purchased and vanadium sample cans purchased through the grant scheme enabled us to make optimum use of this, particularly later on when the powder instruments had their full complement of detectors and powder patterns were being collected much more rapidly.
- High pressure work was anticipated and a 90 tonne hydraulic press was designed and constructed locally.
- The beam line vacated by LONGPOL was again called into service, this time to become an intermediate-Q instrument and a test bed for an area detector, to gain experience before the AUSANS instrument came on line (see below).
- The Low Angle Diffractometer was repositioned onto 4H5B which now had a longer wavelength (2.36 Å) and improved resolution due to the higher monochromatic take-off angle. It had a linear position sensitive detector and was particularly suited to studying biological membranes.
- With commercial interest in mind, the 2TanB single crystal diffractometer was reprogrammed to take texture measurements and produce pole figures.

- A collaborative development of the X166 beam line led to a Laue facility which was used for testing image plate technology for neutrons (essentially a Gadolinium foil converter attached to a photographic plate).
- With overseas assistance, we trialled a small area detector on 2TanA. Although initial trials were successful, follow up measurements with new electronics and improved programming did not give the anticipated benefits, so in the long run the area detector did not see much use.

A major expense was the improvements to the site computing infrastructure, including ANSTOnet. This allowed the instrument control computers to communicate directly with the site computing system without the need for data transfer discs. All the instrument control computers were upgraded over a few years to take advantage of this development.

The planning for AUSANS started in 1987 with an instrument concept totally within the reactor containment building (RCB). Detailed design, however, showed that, to reach a minimum $Q$ of 0.005 Å$^{-1}$, this would not be possible and the decision to go outside the RCB was taken in 1989. To free up the 6H hole, three collimators (4H1, 4H2 and 6H) were redesigned to match their new instruments, a single shield for two powder instruments was designed and constructed. The HRPD from 6H was moved to a new location using the 4H2 position, and the beam selection/optimisation part of AUSANS was installed where the HRPD had been. The wavelength selection was achieved using a pair of multilayers 800 mm long and set at a few degrees to the beam direction. The lateral shift of the beam was fixed and wavelength change achieved by simultaneous angular and longitudinal movement of the multilayers. To achieve an acceptable flux at the sample, the complete length of the beam line was either filled with helium (in-pile collimator) or evacuated. Originally planned for 1989, the changeover took place in July 1991, using a three-month reactor shutdown to remove old instruments, change collimators and install new shielding and instruments. Part of this delay was caused by the difficulty in removing the original 4H2 collimator. During this delay the Medium Resolution Powder Diffractometer (MRPD) upgrade continued with more detector channels being added as funds became available (reaching eight). When the major change occurred, the old gun mount based instrument became a sculpture in the Building 58 courtyard. The AUSANS Instrument itself was located outside the HIFAR Building in a purpose-built room constructed above but totally separate from the Diesel Plant Room. An appropriate valve was installed into the Building Port to allow the evacuated neutron containing pipeline to penetrate the wall. (Fortunately, the original designers of DIDO/HIFAR had allowed for the possibility of taking a beam outside the building). While most of the funds for this instrument came from ANSTO and AINSE sources, the design and construction of the large area detector, which was achieved in-house, was funded by an Australian Research Council (ARC) Large Equipment grant of $250 thousand in 1992.

This was the first of eight ARC grants, under the Research Infrastructure Equipment and Facilities Program (RIEFP), over eight years which provided $1.84 million for neutron instrument capital improvements (Appendix C). The availability of these grants occurred at about the same time as the Research Reactor Review of 1992. This recommended to the government that certain issues needed to be resolved before the decision to build a new reactor was made, one of which was that "there is good evidence of strong and diverse applications of neutron scattering capability in Australian science" [57] (p. XV).

From 1992 to 1999 many substantial instrument upgrades were achieved using these grants.

- The HRPD reached its design limit of 24 detectors with eight detector channels added in 1992 and eight more in 1995, an improved input collimator and a computer upgrade. In its new location the liquid helium based cryostat could no longer be used and a Displex style cryo-refrigerator was also purchased.
- The MRPD reached its design limit of 32 detectors with eight detector channels added in 1993, eight in 1994 and eight in 1996.

- A further upgrade to LONGPOL, in which 97% polarisation was to be achieved using supermirror polarisers, was commenced. This project was in collaboration with the Hahn Meitner Institute, Berlin, and would eventually require some 10,000 silicon wafers to be prepared with special coatings. Potentially this would also allow use of a shorter wavelength with an associated flux increase. LONGPOL slowly acquired the eight polarisers of its final design (1993, 1994, 1996 and 1998 grants) and an upgraded computing system. It also received a closed cycle refrigerator (again from Monash University) for sample environment control with the ability to interchange sample and background positions internally by air pressure.

- 2TanA had been replaced in 1994 and its special area detector electronics was provided in 1995 together with a single crystal furnace.

- Both 2TanA and 2TanB gained new drive electronics and IBM PCs for computer control operation.

- All areas benefitted by the computer hardware upgrades, which were essential to take full advantage of ANSTOnet, and ongoing software support. Programming was especially required to link ancillary devices to the computers for automatic control during experiments and development of a new protocol for time-of-flight data acquisition on LONGPOL.

Over and above the in-house efforts, some significant sample environment equipment was also purchased through the grants scheme; for example, a cryo-magnet, a Langmuir trough for the new reflectometer, both a shear cell and a Couette cell for AUSANS, which also gained a post-doctoral fellow to provide dedicated assistance to users.

A separate outcome from the ASTEC report of 1985 [56], recommending travel funds to allow Australians to perform experiments overseas when appropriate facilities were unavailable in Australia, also met with success. In 1992, an agreement was reached with the Rutherford Appleton Laboratory (RAL) in the UK which, for an appropriate fee, would allow Australians to apply for beam time. ANSTO paid the fee for six years ($2.1 million) and then a further extension was obtained through a RIEFP grant of $400 thousand per year for five years. On top of this were substantial travel monies ($\approx$$200 thousand in the first six years) from the Access to Major Research Facilities Program. As part of the agreement a scientist was based at the RAL for a while as in-kind support and also to learn reflectometry. The main beneficiaries from this funding were the soft matter scientists requiring instruments using cold neutrons, especially small angle neutron scattering and reflectometry.

The improved facilities reversed the downward trend in the number of publications from the HIFAR instruments, which was now on a steady climb (Figure 3).

The status reached by 1998 is given in the excellent correspondent's report in Neutron News [5].

However, more importantly, the government had made a decision to replace HIFAR. If all went according to plan the replacement reactor would be a *multipurpose, swimming pool type reactor, built on the Lucas Heights site with $\approx$3 times increase in source flux, up to 17 thermal and cold neutron instrument locations for a wide range of neutron beam instruments, both at the reactor face and through neutron guides to a neutron guide hall. A cold source will enable the exploration of research areas not previously possible in Australia.* [58] (p.32)

## 5. The Final Years 1999–2007

This was a time of change. The new reactor was on its way, and significant staff effort was already being diverted for tender assessment and contract specification. Committees had been formed, deliberated and produced a list of the preferred initial suite of neutron beam instruments (see below). The instruments were chosen to use the existing expertise both at ANSTO and within Australia and to provide facilities for the Australian community well into the future. Of the instruments still operating at HIFAR, only LONGPOL was originally listed to be moved across to OPAL, although this did not happen. From the initial eight, the 4-circle diffractometer was replaced by the residual stress diffractometer.

Planned Instruments for the OPAL Reactor:

1. High Intensity Powder Diffractometer
2. High Resolution Powder Diffractometer
3. Small Angle Scattering Instrument
4. Vertical Neutron Reflectometer
5. Polarisation Analysis Spectrometer (LONGPOL)
6. 4-Circle Diffractometer
7. Quasi Laue Diffractometer
8. Triple Axis Spectrometer
9. High Energy Resolution Backscattering Spectrometer (by 2010)
10. Amorphous Materials Diffractometer (by 2010)
11. Residual Stress Diffractometer (by 2010 if possible)
12. Radiography (by 2010 if possible)
13. Second SANS (after 2010)
14. Neutron Spin Echo Spectrometer (after 2010)
15. Vertical Neutron Reflectometer (after 2010)
16. 4-Circle Diffractometer for Hot Neutrons (after 2010)
17. Thermal and Cold Neutron Test Beds (after 2010)

The formation of the Bragg Institute in December 2002 was the time when neutron scattering *came-of-age* at ANSTO, no longer an add-on to Materials or Applied Physics Divisions. When formed, the Bragg Institute had two main teams: those involved with the HIFAR instruments (instrument scientists, technicians), and those involved in designing the first eight instruments for the Replacement Research Reactor (OPAL) (lead scientists who got to go around the world to see what everyone else was doing and determine current *World's best practice*, plus engineers and draftsmen seconded from ANSTO Engineering) under the Head (Rob Robinson) and Technical Leader (Shane Kennedy). In typical fashion, the number of people outgrew the building (Building 58) very quickly so a large annex was assembled out the front to house them. When the Australian Synchrotron project staff joined the Bragg Institute even that was not enough and a second annex was added out the back to accommodate them and the increasing number of post-docs based with us. Managing the transition from HIFAR to OPAL started as soon as the replacement reactor was announced. There were seven years in which to get the best out of the HIFAR beams. Two major changes were approved. The lack of a neutron reflectometer had been highlighted in 1998 [5] and it was decided that, with a modest investment, a neutron reflectometer could be constructed using the dogleg beam line of X172. There would be time to get experience with its use before HIFAR shutdown. It achieved five intensity decades of fringes from simple systems and allowed scientists to develop processing programs for the data [59] (highly cited). It was also planned to use it for testing supermirror guide sections during their production for the new era, although this did not actually happen (they were fully tested overseas).

The oldest instrument, the Triple Axis Spectrometer (TAS), was not seeing significant demand. It was decided that with an improved sample stage and the addition of a small area detector it could be converted to a strain scanner (TASS) to give the staff experience in residual stress measurement. This would be an instrument more suited to industrial applications and was strongly supported by research in the Materials Division. Five instruments (HRPD, MRPD, 2TanA, AUSANS and LONGPOL) had recently been upgraded to ensure successful operation until HIFAR shut down. Some noisy detectors on HRPD had been replaced (better quality data), a pneumatic shutter had been installed on MRPD (lower radiation dose to users), AUSANS had some new ancillary equipment (sample changer, Couette cell, and sample temperature control) and improved software, LONGPOL had a supermirror polariser and eight analysers and 2TanA had been given a motor upgrade with new driver software following irreparable failure of the old system.

Ancillary equipment was adequate but becoming unreliable and difficult to maintain (lack of parts and expertise). New ancillaries were purchased, but they had to comply with the new standards to ensure they could be used on the new OPAL instruments. Where possible, if new techniques were being considered for the new instruments, they were tested on the HIFAR instruments before being approved. Examples were: the mounting for focussing monochromators for the powder instruments; trial of a small area detector for single crystal data collection; the use of image plates; and the testing of remote computer control of equipment moving on airpads on a simulated dance floor in B42.

For six years, the HIFAR instruments were run almost non-stop. The users were students, post docs, research scientists, whoever could get approval, getting as much data from the HIFAR instruments before the reactor was shut down. Students with theses to complete got higher priority as it got closer to shut down.

In 2003, both the n-Reflectometer (Figure 4) and TASS (Figure 5) received their first external users. Significant promotional activities were started (the new instruments were planned to have ≈10 times more flux at the sample positions than the current instruments). ANBUG was reinvigorated and given positions on various committees so that external users had a guaranteed say in developments. As a result there was a significant increase in demand from AINSE (proposals up 16%, usage up 28% and 14 new researchers), and ANSTO's Institute of Materials and Engineering Science (IMES) was making good use of TASS. Availability of other instruments was ≈87%.

In 2004, Neutron instrument availability ≈80%.

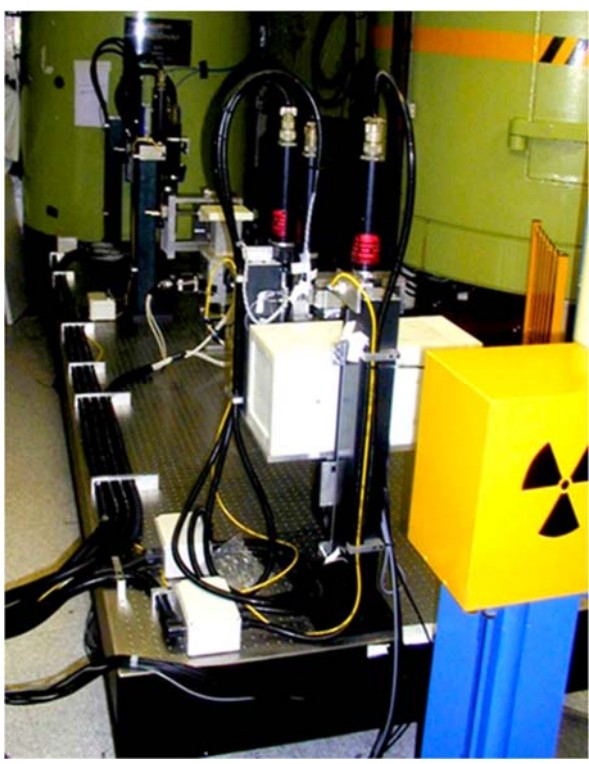

**Figure 4.** Neutron Reflectometer at HIFAR 2004. Australian Centre for Neutron Scattering Photograph archives; © ANSTO used with permission.

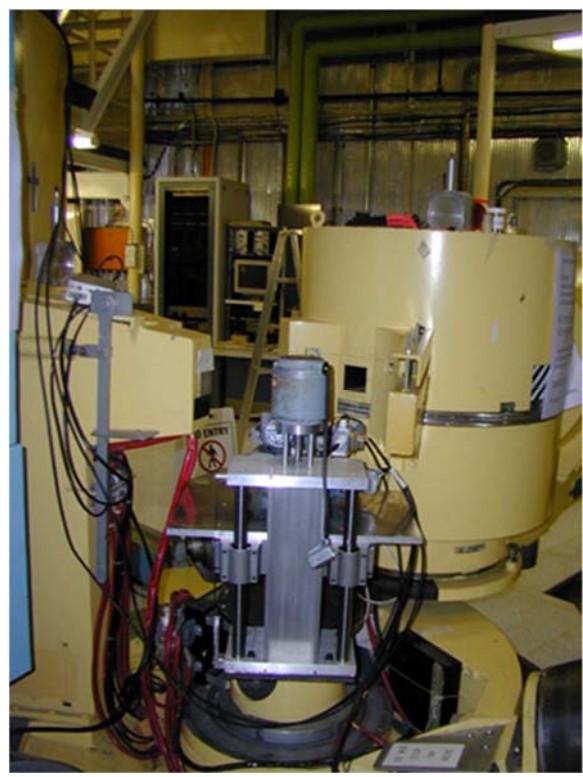

**Figure 5.** The Triple Axis Strain Scanner at HIFAR 2004. Australian Centre for Neutron Scattering Photograph archives; © ANSTO used with permission.

In 2005, there was a significant shift in emphasis to operations only (reduced technical effort as installation of the new instruments started) plus continued efforts to get more demand. Attempts to run LONGPOL at a shorter wavelength were aborted and it ran for the rest of its operations at 3.6 Å. Neutron instrument availability ≈90%. The 7 Tesla cryomagnet, received the previous year was finally commissioned both in the workshop and on MRPD. There was a continued improvement in customer numbers, usage was up 4%, users up by 12 and new users up by 51. A development that took place with the HIFAR instruments was the introduction of a computer based on-line scheduling system. Users could see the schedule from *outside*, and keeping track of who used which instrument for what proposal became much easier. There was significantly greater use of the feedback system and the ratings also improved. To prepare for a much wider range of users (adding ANSTO, ANSTO collaborators and international researchers to the existing AINSE user data base) a review system for non-AINSE proposals commenced.

In 2006, activities at HIFAR started slowing down. LONGPOL ceased operation in March and 2TanA in June. All the rest ran right up to the end.

The end of the HIFAR era came on 30 January 2007, when the Minister for Education, Science and Training Julie Bishop pressed the button to initiate HIFAR's final shut down.

There was an overlap of a few months during which both reactors operated simultaneously. The first data from the new HRPD (ECHIDNA) were collected in December 2006 and the first paper was published a year later. The contrast between the cramped conditions in the single HIFAR building containing the reactor and all seven neutron beam instruments; (neutron scattering equipment had been allocated 58% of the main floor level + an added mezzanine floor level for instrument control and ancillary equipment when not in use;) and the huge warehouse-like initially empty guide hall was staggering. No longer would the instrument shielding design be dictated by the ability to get it into position (AUSANS being the last and hardest to get in). The new instruments were designed for optimum performance including low radiation background levels.

There was also a nine-month decommissioning plan for all the neutron instruments inside HIFAR. As soon as practical after the shut down all re-useable items were removed from the HIFAR building and transferred to storage. The 2TanA Instrument found a new home in the Power House museum. Much later, months after the shutdown, when components had cooled, I went back into the HIFAR RCB with the active handling crew and radiation survey worker to remove all the useable monochromators—large single crystals—from their shielding and transfer them across to storage in the reactor beam hall. Some of them are now in use on OPAL instruments.

## 6. Epilogue

In the 40 years I worked at Lucas Heights, neutron scattering grew from five single detector un-computerised instruments (data accumulated on rolls of paper tape which had to be taken to the site Main Frame computer for processing) to seven highly sophisticated, computer controlled instruments using single detectors, multidetectors or area detectors as appropriate, with an approximate 16-fold increase in data collection rates.

Many well-known scientists, Hugo Rietveld, Trevor Hicks, Alan Hewat, Sax Mason, John Parise, Oscar Moze, Andrew Wildes and Dimitri Argyriou, to name a few, started their scientific careers at HIFAR. Achievements in neutron scattering from using neutron beams from HIFAR are given below.

From 1958 to 2006:

- 740 refereed publications (many with >150 citations, Appendix A)
- 2 DSc theses
- 179 PhD theses
- 17 MSc theses
- 10 Honours theses

Data were still being collected at shutdown and these figures are undoubtedly an underestimate.

Scientific awards:

David Syme Research Prize:

- Terry Sabine, Arthur Pryor, Brian Hickman and Terry Walker

Three Lucas Heights Scientific Society prizes:

- Dr Chris Howard
- Dr John Taylor
- Dr Gerard Gadd

AINSE Gold medal winners for excellence in science 1992–2006:

Senior researchers:

- Dr Trevor Hicks, 1997
- A/Prof Brendan Kennedy, 2003
- Dr Shane Kennedy , 2005

Post graduate students:

- Ismunandar, 1998
- Dr Darren Goossens, 1999
- Dr Gianluca Paglia, 2004
- Mr John Daniels, 2006

I have been asked if there was a scientist who had influenced my scientific career and initially I was stumped for an answer. Now I know that HIFAR has played that role. I adapted my life, as my family know too well, to ensure excellent usage of the neutron beam facilities. The following poem summarises my mentor—HIFAR - five initials, five decades, five attributes

**H**igh on the hill, clad in white, you lived.
**I**nspiring us, through many challenges, to reach our goals.
**F**ocussing our lives, with plans and trips, to keep you well.
**A**chievements from your life, both known and yet to come, encompass much.
**R**eactor on the hill, shrouded in white, still.

All of this would not have been achieved without the unstinting effort of the reactor operations group and other on-site support over the years while HIFAR operated and still continue in the decommissioning phase. We coexisted—more as a family than a team—so the following poem is for them.

### HIFAR—Our Queen

Carefully, willingly, lovingly
She has been tended
by an ever changing myriad of workers,
from far and near,
into her 50th year.

Now,
still carefully, but unwillingly and with heavy hearts
the surviving workers
take Her apart, piece by agonising piece until,
as life dictates She must,
HIFAR is returned to dust.

**Acknowledgments:** I wish to thank ANSTO for continuing access to the Lucas Heights site and its facilities. In particular, I have been privileged to be given unlimited access to the ANSTO library, HIFAR archives, and original instrument log books. I also thank AINSE for access to all the AINSE Annual reports and Council meeting notes. Finding out what took place between 1960 and my arrival in 1967 took many months.

**Conflicts of Interest:** The author declares no conflict of interest.

## Appendix A

Most highly cited papers from HIFAR (at November 2016 using Google Scholar).

Hill, R.J.; Howard, C.J. Quantitative phase analysis from neutron powder diffraction data using the Rietveld method. *J. Appl. Crystallogr.* 1987, *20*, 467–474. (915 citations).

Nelson, A. Co-refinement of multiple contrast neutron/X-ray reflectivity data using MOTOFIT. *J. Appl. Crystallogr.* 2006, *39*, 273–276. (487 citations).

Howard, C.J.; Sabine, T.M.; Dickson, F. Structural and Thermal parameters for Rutile and Anatase. *Acta Crystallogr.* 1991, *B47*, 462–468. (477 citations).

Howard, C.J.; Hill, R.J.; Reichert, B. Structures of $ZrO_2$ polymorphs at room temperature by High Resolution Neutron Powder Diffraction. *Acta Crystallogr.* 1988, *B44*, 116–120. (396 citations).

Kisi, E.H.; Elcombe, M.M. U-parameters for the Wurtzite structures of ZnS and ZnO using powder neutron diffraction. *Acta Crystallogr.* 1989, *C45*, 1867–1870. (370 citations).

Bolzan, A.A.; Fong, C.; Kennedy, B.J.; Howard, C.J. Structural Studies of Rutile Type Metal Dioxides. *Acta Crystallogr.* 1997, *B53*, 373–380. (319 citations).

Howard, C.J. The approximation of asymmetric neutron diffraction peaks by sums of gaussians. *J. Appl. Crystallogr.* 1982, *15*, 615–620. (287 citations).

Rietveld, H.M.; Maslen, E.N.; Clews, C.J.B. An X-ray and Neutron Diffraction Refinement of the Structure of *p*-Terphenyl. *Acta Crystallogr.* 1970, *B26*, 693–706. (209 citations).

Hill, R.J.; Flack, H.D. The use of the Durbin-Watson *d*-statistic in Rietveld analysis. *J. Appl. Crystallogr.* 1987, *20*, 356–361. (198 citations).

Elcombe, M.M.; Taylor, J.C. A neutron determination of the crystal structure of thiourea and deuterated thiourea above and below the ferroelectric transition. *Acta Crystallogr.* 1968, *A24*, 410–420. (180 citations).

Harada, J.; Pedersen, T.; Barnea, Z. X-ray and Neutron Diffraction Study of Tetragonal Barium Titanate. *Acta Crystallogr.* 1970, *A26*, 336–344. (174 citations).

Kennedy, B.K.; Hunter, B.A.; Howard, C.J. Structural and bonding trends in Tin Pyrochlore Oxides. *J. Solid State Chem.* 1998, *130*, 58-65. (165 citations).

Elcombe, M.M.; Pryor, A.W. The Lattice Dynamics of Calcium Fluoride. *J. Phys. C* 1970, *3*, 492–499. (152 citations).

Cadogan, J.M.; Li Hong Shuo;, Margarian, A.; Dunlop, J.B.; Ryan, D.H.; Collocott, S.J.; Davis, R.L. New Rare-earth Intermetallic Phases $R_3(FeM)_{29}X_n$: (R=Ce, Pr, Nd, Sm, Gd; M=Ti, V, Cr, Mn; and X=H, N, C). *J. Appl. Crystallogr.* 1994, *76*, 6138–6143. (143 citations).

**Appendix B**

Instrument placards 1969 and 1971 [60].

<div align="center">

**TRIPLE AXIS SPECTROMETER 1971**
(variable incident neutron energy)

</div>

This instrument is used for studying the vibrations of atoms in crystals. From these studies, the forces which act between the atoms can be found. Because the neutrons required have to be scattered by three crystals, the number which reach the counter is very low. The shielding which absorbs unwanted neutrons is therefore very large (total weight 22 tons). It is fully automatic. The user controls the three crystal orientations and the three scattering angles from the teleprinter through the PDP-8 computer opposite.

<div align="center">

**LONG WAVELENGTH MECHANICAL MONOCHROMATOR**

</div>

A beam of thermal neutrons emerging from the reactor is composed of neutrons with a wide range of energies. A monochromatic beam of neutrons (i.e., neutrons of a fixed energy or wavelength) may be selected from the emerging beam by means of a crystal monochromator for wavelengths up to 5 Angstroms, or by a mechanical monochromator at wavelengths greater than about 5 Angstroms. A mechanical monochromator consists of a motor driven slotted rotor which is set at an angle to the incident beam. The velocity, and hence the wavelength of the neutrons emerging from the rotor slits will be proportional to the angular velocity of the rotor. Specimens are cycled in and out of the monochromatic beam to make allowance for fluctuations in the emerging beam intensity due to variations in reactor power.

<div align="center">

**LONG WAVELENGTH NEUTRON SCATTERING**

</div>

When a beam of neutrons passes through a sample of material suitable for use as a moderator, attenuation of the beam is caused almost entirely by Bragg scattering. If a long neutron wavelength is chosen such that the Bragg condition is not satisfied by any set of planes in the crystal, there is very little attenuation. If the sample is irradiated by fast neutrons, atoms are removed from lattice sites

causing imperfections in the crystal lattice. These imperfections give rise to extra attenuation of the long wavelength beam. Examination of the variation in attenuation with wavelength before and after irradiation gives information about the distribution and number of defects.

## SMALL ANGLE MAGNETIC SCATTERING

Ferromagnetic or anti-ferromagnetic ordering in a crystal is controlled by long range co-operative forces. Information about the nature of these forces can be obtained from the way in which their thermodynamic fluctuations (magnons) scatter neutrons. This scattering is strongest in the forward direction because of the dependence of the intensity of magnetic scattering on scattering angle. The apparatus is being used to investigate magnetic coupling in some ferromagnetic iron-nickel alloys. The direction of the applied magnetic field is changed to enable nuclear scattering, which is independent of field, to be subtracted.

## POLARISED NEUTRON DIFFRACTOMETER 1971

This instrument is used for studying certain features of magnetic structures. It uses a beam of neutrons which have been polarised so that the magnetic moments of the neutrons are aligned vertically. A 1/2Mc/s R.F. coil may be used to reverse the spin direction of the beam incident on the sample. The sample is held, either at room temperature or in a cryostat or furnace, between the poles of a high field electromagnet. Scattering of the neutrons is recorded in the tilting counter. The positions of the counter and sample, and the state of the neutron spin flipper, are controlled by a PDP-8L computer.

## FOUR CIRCLE SINGLE CRYSTAL DIFFRACTOMETERS 1969

These instruments are used for accurately determining the positions and thermal vibration amplitudes of the atoms in the single crystal specimen. Neutrons of wavelength 1 Angstrom ($10^{-8}$ cm) are Bragg reflected from crystal planes, and the integrated intensities of these reflections are measured by step scanning the counter and crystal together. A structure determination might require 200 to 2000 such intensity measurements from different planes, each taking 10 to 100 min to measure. For each measurement, the crystal must be oriented using the inner two circles ($\chi$ and $\phi$) so that the desired plane is vertical, and reflects the horizontal beam into the counter. The crystal and counter angles ($\theta$ and $2\theta$) must then be set to satisfy Bragg's law, $\lambda = 2d_{hkl} \sin\theta$, where $\lambda$ is the wavelength of neutron, $d_{hkl}$ = interplanar spacing. The intensities of the reflections are corrected for various geometrical and experimental factors, and then processed by large computer programs to derive the structural parameters. The instrument here is manually operated—the crystal must first be aligned correctly with a known direction along the $\phi$ axis. Setting angles must be calculated and set by hand for each reflection in turn. Step scanning is then automatic, the results being punched on paper tape for processing by the site computer.

## COMPUTER CONTROLLED 4-CIRCLE DIFFRACTOMETER 1969

This is similar to the manual machine on the left, but it is completely automated, using the PDP-8 computer situated on the gallery above. The user controls all operations from the teleprinter below. It is not necessary to accurately align the crystal—the computer can centre several reflections automatically using the $\chi$, $\Omega$ and $2\theta$ motions, and from their positions, the orientation of the crystal can be calculated exactly. The user can then request the computer to calculate the setting angles for any given reflection, drive the motors to these angles, and measure the reflection. A large series of such reflections can be measured without manual intervention.

## COMPUTER CONTROLLED 4-CIRCLE DIFFRACTOMETER 1971

This instrument is one of two fully automatic diffractometers controlled by the PDP-8 computer situated on the gallery above. The diffractometers operate completely independently of one another, in a time shared mode, using their own storage areas in the computer but sharing the same

program. The user controls the operations of the diffractometer from its associated teleprinter. These diffractometers are used for measuring Bragg intensities or for measuring the intensity of diffuse scattering (they need not both be doing the same thing), and small furnaces and cryostats are available to heat or cool the crystal sample.

### POWDER DIFFRACTOMETER

This instrument is used to record the neutron diffraction pattern of substances which can be obtained only as powders. In a powder, the Bragg condition will be satisfied for all reflections simultaneously since the grains have a random orientation. Only a small fraction of the specimen contributes to each diffraction line so the intensity is low. This technique is used principally to investigate the magnetic structure of materials. Work can be carried out at temperatures down to 4 K ($-269$ °C) and in magnetic fields up to 20 kilogauss.

**Appendix C**

**ARC RIEFP grants for capital investment**

These came under a government initiative to boost scientific infrastructure. They were submitted by AINSE with University and ANSTO support both in kind and in cash. All values are in Australian \$. The list below is from the applications. We did not always get as much as we requested.

1992—ARC large equipment grant \$250 thousand for area detector for AUSANS.

1992—Infrastructure grant for neutron scattering \$221 thousand –\$80 thousand 8 detector channels for HRPD, \$90 thousand for a cryo-refrigerator, and \$50 thousand for a programmer.

1993—\$300 thousand request, \$80 thousand for 8 detecting channels for HRPD, \$20 thousand for control computer upgrade, \$30 thousand for supermirror spin polarisers, and \$170 thousand for staff and consumables.

1994—\$275 thousand –\$80 thousand for upgrading MRPD and \$75 thousand for LONGPOL, \$60 thousand towards continuing modernisation of instrument control systems, \$43 thousand for staff support.

1995—requested \$300 thousand awarded \$180 thousand–\$80 thousand to expand HRPD channels from 16 to 24, and new high resolution input collimator, \$70 thousand for special electronics for area detector and a single crystal furnace for 2TANA, \$30 thousand for staff support.

1996—\$225 thousand –\$80 thousand 8 detector channels for MRPD (to reach design limit of 32), \$75 thousand for more supermirrors for LONGPOL, \$45 thousand staff support, \$20 thousand for shear cell for AUSANS.

1998—\$150 thousand –\$50 thousand for staff support, \$70 thousand for coating silicon wafers, \$30 thousand for monochromator crystals.

1999—requested \$320 thousand awarded \$240 thousand –probably \$60 thousand for staff support, \$130 thousand for a cryo-magnet, \$50 thousand for Langmuir trough for neutron reflectometer.

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
