# Peer review of "Neutron Scattering at HIFAR—Glimpses of the Past"

_qubs, doi:10.3390/qubs1010005_

Round 1

Reviewer 1 Report

Please see the attached PDF file of comments. I have recommended that this paper be published with the following minor corrections.

79Because HIFAR would be a carbon copy...

85The 4H2 collimator... 

90The first instrument, a long wavelength spectrometer, was installed on 4H1 in April 1960, but I think it was never used as a spectrometer there.

145 Fourier maps were used to locate the N and O atoms in the 0kl and hk0 planes..

146An AINSE postdoctoral fellow... name ?

280The strategy for improving data reliability is interesting. Who introduced that ? [37] ?

Author Response

Indeed it would be nice to have a simple plan of the HIFAR beam tube layout. Unfortunately one does not exist. In the interests of accuracy I am not going to try and create one. That is why I used the list from the OPEN DAY 1971 booklet, which is at least recorded and accurate.

The version of table 1 had been corrupted by early editing. The original table is as the reviewer suggests it should be. The original instrument names have been maintained so that anyone using this information in the future will not be misled.

The bulk of this paragraph has been added near the end of section 2.

As pointed out in footnote 4 there are known issues with the HIFAR publication list, but it is the only one I have. Did the publications really decline? or is it that requests to the users to give us their publication lists fell on deaf ears. Were the journals expecting more quality over quantity? ie one structure was no longer enough. About this time there was a steady increase in publications by Australians from data collected overseas. The text has been modified slightly.

Expansion did start in 1986 with AUSANS and the upgrade of both the HRPD and the MRPD although the fruition did not occur until 1991. This tided us over until the grant monies were achieved. ‘tough times’ has been deleted, the text modified slightly, and a comment about revenue included. We never lost any staff, they were just paid for from different sources.

‘expected to occur within the next 5 years’ has been deleted.

I agree with the reviewer that the RAL access could be described as ‘curious’. As an historian I am reporting what happened not why. No change to the text is warranted.

The epilogue would not be complete without both poems. Please leave as is.

Appendix B. While agreeing that this information is unnecessary for the professional reader, it is included to point out to the younger generation that we could not produce fancy posters. A comment has been added in the text.

Reviewer 2 Report

This is an historical account of the research and development that took place in the field of neutron scattering at the HIFAR reactor at ANSTO from 1958 through to the closure of the reactor in 2007. The author, who started working at the reactor in 1967, has looked back through the records to give an account from the beginning. 

There is an enormous amount of information in this document in how the instruments and their usage changed over the 49 years of the reactor's life. The author has noted the important changes, such as the increasing use of the instruments by University users, and has listed the important papers published from the work, as well as how the instrument suite evolved. Of course, changes in the computing business also played a key role, and the author has emphasised this aspect, as well as modern detector technologies opening new horizons. It is one of the most complete accounts of any such project in the world, the only scattering site so well documented over such a long time is NIST outside Washington DC. This long perspective on HIFAR allows some clear lessons for the future to be drawn.

In confining remarks to the HIFAR reactor, I think the author might have slightly played down the importance of the new reactor OPAL, which is now operating at the same site. For example on p. 17 “The Final Years 1999-2007”, the author does not give perhaps the clearest account of how the performance of the instruments at HIFAR must have undoubtedly influenced the choice of instruments for OPAL. For example, it would be useful to have a table to show how many instruments were moved from HIFAR to OPAL, and how many new instruments were on the drawing board for OPAL, and how many did not continue. In some sense, this gives the historical continuity between the two reactors, and shows the continual maturing of the neutron scattering program in Australia – one can truly say that the work at OPAL builds on the pioneering work at HIFAR. A Table might make this clearer.

On one subject I think the author needs to add a few more details. Some neutron instruments perform research that is close to applications; examples are strain scanners, reflectometers, and small-angle scattering instruments. The author writes on p. 15 (line 473) "The Government also insisted that the Organisation bring in real revenue to supplement the appropriation funds. If the revenue targets were not met then staff would be cut – tough times." Now this refers to the period about 1990. In fact these policies were first tried at Harwell starting in the late 1960s, and have been implemented at other places as well. They have generally failed to bring in significant revenue, although (with strain scanning, for example) they have succeeded in getting the method accepted by engineers. However, the author of this work never returns to this subject again. Historians will be interested in the results found for Australia, so I think some conclusions should be made in the view of historical assessment. This need not be more than a paragraph to give the results, and their influence on continuing this policy or not, as the case might be.

I congratulate the author on a very fine piece of research, some of it through the old archives, and of providing a document of historical interest as well as one that has lessons for the future of large facilities, especially in countries like Australia.

Author Response

I have included the list of initial instruments at OPAL, with appropriate text.

. ‘tough times’ has been deleted, the text modified slightly, and a comment about revenue included. We never lost any staff, they were just paid for from different sources.

Round 2

Reviewer 2 Report

I have looked at the changes and they are satisfactory for me.